# Euendolithic Infestation of Mussel Shells Indirectly Improves the Thermal Buffering Offered by Mussel Beds to Associated Molluscs, but One Size Does Not Fit All

Alexia M. Dievart [1,*], Christopher D. McQuaid [1], Gerardo I. Zardi [1,2], Katy R. Nicastro [1,3,4] and Pierre W. Froneman [1]

1. Coastal Research Group, Department of Zoology and Entomology, Rhodes University, Grahamstown 6139, South Africa
2. UMR 8067 BOREA–Laboratoire Biologie des Organismes et Ecosystèmes Aquatiques, CNRS, MNHN, UPMC, UCBN, IRD-207, University of Normandie, UNICAEN, CS 14032 Caen, France
3. UMR 8187–LOG–Laboratoire d'Océanologie et de Géosciences, University of Lille, CNRS, University Littoral Côte d'Opale, F-59000 Lille, France
4. CCMAR–Centro de Ciencias do Mar, CIMAR Laboratório Associado, University of Algarve, Campus de Gambelas, 8005-139 Faro, Portugal
* Correspondence: alexia.dievart@hotmail.fr; Tel.: +27-72-582-9187

**Abstract:** Mussel beds form important intertidal matrices that provide thermal buffering to associated invertebrate communities, especially under stressful environmental conditions. Mussel shells are often colonized by photoautotrophic euendoliths, which have indirect conditional beneficial thermoregulatory effects on both solitary and aggregated mussels by increasing the albedo of the shell. We investigated whether euendolithic infestation of artificial mussel beds (*Perna perna*) influences the body temperatures of four associated mollusc species during simulated periods of emersion, using shell temperature obtained via non-invasive infrared thermography as a proxy. Shell temperatures of the limpet *Scutellastra granularis* and the chiton *Acanthochitona garnoti* were higher in non-infested than infested mussel beds during simulated low tides under high solar irradiance and low wind speeds. However, this was not the case for the limpet *Helcion pectunculus* or the top shell *Oxystele antoni*. Morphological differences in mollusc shape and colour could, in part, explain this contrast between species. Our results indicated that endolith-induced improvements in humidity and temperature in mussel beds could benefit associated molluscs. The beneficial thermal buffering offered by euendolithic infestation of the mussel beds was effective only if the organism was under heat stress. With global climate change, the indirect beneficial effect of euendolithic infestation for invertebrate communities associated with mussel beds may mitigate intertidal local extinction events triggered by marine heatwaves.

**Keywords:** infrared thermography; ecosystem engineers; *Perna perna*; ecosystem functioning; desiccation stress; heat stress; parasitism; mutualism; invertebrate communities

## 1. Introduction

Heatwaves are discrete periods of prolonged, abnormally warm air (i.e., atmospheric heatwave) or seawater temperature (i.e., marine heatwave) at a particular location [1]. The frequency, duration, and intensity of atmospheric and marine heatwaves have increased since the pre-industrial era [2–4] and are expected to increase with global climate change (GCC), with many parts of the coast and ocean reaching a state of near-permanent heatwave by the end of the century [5–7]. These events are triggered by anomalous weak wind speeds and increased solar radiation, amongst other environmental factors [8,9]. Since organisms suffer more from extremes than slow changes in their environment [10], heatwaves have been implicated in large-scale shifts in species location, phenological changes, changes in ecosystem structure, and elevated levels of mortality in coastal ecosystems [11–14].

This is particularly true for rocky intertidal shores, where temperature and desiccation stress are extremely high during aerial exposure. This is especially the case on sunny days with little to no overcast and low to no wind [15]. However, the effect is highly variable at small scales due to the complex topography of these shores [11,16,17] and at larger scales due to the timing of tides and local weather patterns [15,18]. Intertidal organisms already live at, or close to, their upper thermal tolerance limits and are thus more susceptible to local extinction events [11,19,20]. However, intertidal organisms can display behavioural thermal adaptations to reduce their temperatures, such as 'mushrooming' [21], changing the orientation of the shell [22], or seeking thermal refuge in a more favourable microhabitat [23,24], which are only effective if the organisms are under stress [22].

On rocky shores, mussel beds form an important intertidal habitat that mitigates both temperature and desiccation stress, creating a suitable microhabitat for many species, particularly under extreme environmental conditions [25–28]. Not only do these ecosystem engineers (*sensu* [29]) provide a thermal refuge, but the properties of the mussel bed influence the quality of the habitat they provide [30–32]. Worldwide, mussel shells are targeted by photoautotrophic euendolithic microorganisms, which can indirectly modify the mussel phenotype, thus influencing the thermal buffering provided by mussel beds [31,33,34]. Photoautotrophic euendoliths refer to a heterogeneous group of microorganisms (i.e., cyanobacteria, chlorophytes, and rhodophytes) that live and actively bore into relatively soluble substrates, such as phosphate and carbonate substrates (e.g., coral skeletons, bivalve shells) [35–37]. Present essentially anywhere there is sufficient light for photosynthesis and a suitable substratum to bore into, photoautotrophic euendoliths readily infest marine calcifiers (i.e., corals, bivalves, crustose coralline algae), with both negative and positive effects [38]. In addition to its negative sub-lethal and lethal effects on mussels [38–40], euendolithic infestation causes a distinctive discolouration of the mussel shell as a by-product of its corrosive activity, thereby increasing its albedo [33,34]. Since they reflect more light, infested mussels display lower body temperature and greater survival rates than non-infested mussels during heatwaves [34,41]. This beneficial effect of euendolithic infestation extends to both infested and non-infested neighbouring mussels but is restricted to periods of intense heat and desiccation stress [31,41,42]. It is anticipated that GCC and ocean acidification will coincide with increased rates, prevalence, and severity of euendolithic infestation of mussels [38]. Investigating the potential impacts of euendolithic infestation of mussel beds on the thermal buffering they provide to associated species is critical to predicting how intertidal ecosystems will react to GCC in the future.

To monitor the ecological responses of organisms to GCC, ecologists need to use tools and methodologies that can capture the actual conditions experienced by targeted species within their natural microhabitats, often at small scales [43,44]. Infrared thermography (IRT) is a non-invasive tool for the rapid detection and measurement of small-scale temperature variability in situ without disturbing the animals' behaviour or thermoregulatory capacities [17,45]. Previously restricted to terrestrial ecology, IRT has been successfully employed in intertidal ecosystems to investigate the role of small-scale temperature variability on the physiology and ecosystem functioning of intertidal communities [11,22,46,47]. Shell temperatures of intertidal molluscs measured by IRT have been shown to be strongly correlated with internal body temperatures [46,48,49]. Body temperatures of intertidal invertebrates are determined by heat fluxes towards and from the organism [50,51], which are dependent on the interaction between climatic heat sources at macroscales (e.g., air and water temperatures; [11,26]), non-climatic heat sources at the niche level (e.g., solar irradiance and re-radiation from the substratum; [52]), and biotic factors at the organismal level (e.g., shell colour, morphology and size, behaviour, selection of thermally favourable microhabitats; [22,51]). Non-climatic heat sources appear to have primary control over body temperatures of intertidal ectotherms [52], along with substratum temperature for organisms with a large foot that maintains a conductive, direct contact with the substratum (e.g., limpets, snails, chitons, barnacles; [46,48,50,53]).

In this context, the present study investigated whether euendolithic infestation of mussel beds alters, during aerial exposure, the body temperatures and desiccation stress of selected molluscs known to inhabit mussel beds on South African rocky shores. We hypothesized that: (i) the body temperatures of key intertidal species would be lower, and warming rates slower, in beds formed by endolith-infested mussels than non-infested individuals, (ii) similarly, loss of water by key intertidal species would be lower, and their desiccation rates slower, in endolith-infested beds, and (iii) this would only be true under high temperature and desiccation stress (i.e., at midday on days with high solar irradiance and low wind speed).

## 2. Materials and Methods

### 2.1. Focal Species and Experimental Setting

We selected four key intertidal invertebrate species: the limpets *Scutellastra granularis* (Linnaeus, 1758) and *Helcion pectunculus* (Gmelin, 1791), the chiton *Acanthochitona garnoti* (Blainville, 1825), and the top shell *Oxystele antoni* D.G. Herbert, 2015. These mobile molluscs were specifically selected as: (i) they are able to move between microhabitats during emersion and are commonly found in mussel beds, (ii) they have a rather large foot that maximizes conductive contact with the substratum, and (iii) they suffer differently from high solar irradiance because of differences in shell pigmentation and texture, as these both affect the absorption of solar radiation [54]. Invertebrates were collected on the rocky shores of Port Alfred (Shelly Beach—33°36′49.0′′ S, 26°53′20.7′′ E). Prior to each experiment, invertebrates were kept in seawater (20–22 °C), with rocks colonized by microalgae collected from the rocky shores of origin, under a 12 h/12 h light/dark regime, for 48 h.

They investigated molluscs present morphological differences in shape and colour. On the one hand, the chiton *Acanthochitona garnoti* and the limpet *Scutellastra granularis* (heavily eroded in this study) are both light in colour, with a large foot relative to their height, and display little to no shell features. By contrast, the limpet *Helcion pectunculus* and the top shell *Oxystele antoni* are both dark in colour. While *H. pectunculus* has a similar shape and foot size to *S. granularis*, its shell is even more heavily sculpted. Conversely, *O. antoni* has a coiled smooth shell with a smaller aperture and can retract its foot inside its shell when stressed.

Artificial mussel beds were created from shells of the brown mussel, *Perna perna* (Linnaeus, 1758), a dominant ecosystem engineer on rocky shores on the South and East coasts of South Africa [55,56]. Artificial mussel beds consisted of two treatments: (a) 100% non-infested *P. perna* (*n* = 3) and (b) 100% infested *P. perna* (*n* = 3). Mussels (shell length: 4–5 cm) were collected on the rocky shores of Port Alfred (Shelly Beach—33°36′49.0′′ S, 26°53′20.7′′ E) and Kasouga (Ship Rock—33°38′54.3′′ S, 26°45′31.6′′ E), with care taken to sample mussels displaying either low (stage A-B, for the purpose of this study classified as 100% non-infested) or high (stage D-E, classified here as 100% infested) levels of euendolithic infestation [33,57]. After collection, the mussel shell was cleaned of its epibionts, while the soft tissue was replaced with a silicone sealing compound (Bostik Marine Silicone Sealant) and left to dry at air temperature for at least 48 h before use. The sealant is known to mirror the temperature of mussel tissue closely in these circumstances [58]. Between 70 and 80 biomimetic mussels were then arranged vertically, with the umbo towards the substrate to mimic their natural position on the shore, in each basket (diameter: 20 cm) made of semi-rigid, white PVC net (mesh size: 4 cm) to create artificial mussel beds [31,34,59].

The series of experiments described below were conducted on a flat roof of the Zoology and Entomology Department of Rhodes University, in Grahamstown, during austral summer (between April and May 2021 and in February 2022), on sunny days, with little to no overcast and low wind speeds (Table 1) and high sun elevation (10 am–3 pm) to maximize the likelihood of extreme temperature and desiccation stress.

**Table 1.** Meteorological conditions for our series of experiments investigating temperature stress with the corresponding infaunal species investigated.

| Date | Infaunal Species | Size Range (mm) [1] | UV Index [2] | Air Temperature [3] (°C) | Humidity [3] (%RH) | Pressure [4] (×10³ Pa) | Wind Speed [4] (km.h⁻¹) | Wind Direction [4] |
|---|---|---|---|---|---|---|---|---|
| 13 April 2021 | *Helcion pectunculus* *Oxystele antoni* | 18–23 13–16 | 7 | 26.11–27.78 | 38–43 | 96.2–96.5 | 7.88–8.69 | S-E |
| 14 April 2021 | *Scutellastra granularis* | 20–30 | 6.5 | 22.78–25.00 | 45–55 | 96.2–96.5 | 13.68 | S-W |
| 15 April 2021 | *Helcion pectunculus* *Oxystele antoni* | 18–25 13–16 | 6 | 22.22–22.78 | 41–44 | 95.8 | 16.90–19.79 | S-W |
| 18 April 2021 | *Scutellastra granularis* | 20–30 | 5.5 | 25.00–27.78 | 11 | 95.8 | 19.47–21.24 | N-W |
| 9 February 2022 | *Acanthochitona garnoti* | 19–25 | 13 | 25.00 | 47–49 | 95.2 | 19.47–21.24 | S |
| 10 February 2022 | *Acanthochitona garnoti* | 19–25 | 12 | 25.00–26.11 | 43 | 95.5 | 14.00–15.45 | S |

[1] Shell length range for *Scutellastra granularis* and *Helcion pectunculus*, shell diameter range for *Oxystele antoni*, and body length range for *Acanthochitona garnoti*. [2] https://www.weatheronline.co.uk (accessed on 12 February 2022). [3] Direct measurements using a thermocouple and a hygrometer for air temperature and relative humidity, respectively. [4] https://world-weather.info/forecast/south_africa/grahamstown/ (accessed on 12 February 2022).

Prior to each experiment, artificial mussel beds were kept in the dark overnight in seawater (20–22 °C) to ensure their exposure to the same environmental conditions (temperature and light). After emersion, artificial mussel beds were positioned in pairs (i.e., one non-infested bed and one infested bed) on previously soaked bathmats to simulate the matrix of a natural mussel bed, which is made up of byssal threads, pebbles, broken shells, and sediment, and retains moisture during aerial exposure. This also avoided direct contact with the underlying concrete. Early trials demonstrated that, without the soaking process and the use of mats, artificial mussel beds would dry out in about 20 min, and the infaunal specimens subsequently died of desiccation after about 30 min.

All statistical analyses were performed in R [60].

### 2.2. Body Temperature Assessment

At the beginning of the experiment, 4–8 randomly chosen specimens of the invertebrate species investigated were transferred from their holding tank to each artificial mussel bed. Artificial mussel beds and their infauna were then exposed to the sun for 90 min, during which thermal images were captured every 5 min. All infrared pictures were taken at the height of approximately 1.5 m from ground level, following a consistent order, using a Testo 882 (Testo AG, Germany) hand-held infrared camera with emissivity values set at 0.95 [45], a thermal sensitivity of 50 mK, and an accuracy of ± 2 °C or ± 2% of reading, whichever was greater. The experiment was run twice on different dates for each invertebrate species and concurrently in the case of *Oxystele antoni* and *Helcion pectunculus* (Table 1).

All infrared images were analyzed using IRSoft 4.8. Within each bed, all infaunal specimens, and five random mussels, were selected at each time interval, and their shell temperatures were recorded using the one point-measure tool targeting the middle of each infaunal and mussel specimen.

### 2.3. Desiccation Assessment

Our series of desiccation experiments were conducted on the following dates: 18 April 2021 (*Oxystele antoni*), 1 May 2021 (*Acanthochitona garnoti*), 2 May 2021 (*Scutellastra granularis*), and 17 May 2021 (*Helcion pectunculus*).

At the beginning of the experiment, 15 randomly selected specimens of each infaunal species were transferred from their holding tank to each artificial mussel bed. Artificial mussel beds and their infauna were then exposed to the sun for 90 min. At selected time intervals (i.e., at the start of the experiment, then after 15, 30, 60, and 90 min), three random individuals for each species were selected from each mussel bed and stored in separate sealed tubes in the shade. At the end of the experiment, each specimen was weighed (wet weight), then dried at 60 °C for 48 h, and weighed again (dry weight). The experiment was run only once for each invertebrate species.

*2.4. Data Analyses*

For both experiments, data did not fit the assumptions of normality and homogeneity of variances. We used a series of Generalized Additive Models (GAMs) to assess the effect of euendolithic infestation overtime on the shell temperatures of both biomimetic mussels and infaunal invertebrate species, as well as on the wet weight of infaunal invertebrate species. GAMs were selected to model these data due to the non-linear relationships present. For the body temperature assessment, either biomimetic mussel or infaunal species shell temperatures at each time step were specified as a continuous response variable and infestation status (infested/non-infested) as a categorical parametric fixed effect. For the desiccation assessment, the wet weight of individual invertebrates was log-transformed and specified as a continuous response variable and infestation status (infested/non-infested) and invertebrate species as categorical parametric fixed effects. A unique identifier code was given to each biomimetic mussel (for the body temperature assessment) or infaunal specimen measured (for both assessments). For all biomimetic mussels and infaunal invertebrate species (except *Acanthochitona garnoti*), a non-linear relationship between infestation and shell temperature was specified using a smoothed cubic regression spline with $k = 4$ knots. For *Acanthochitona garnoti* and the corresponding biomimetic mussels, a non-linear relationship between infestation and shell temperature was specified using a smoothed cubic regression spline with $k = 6$ knots. A non-linear relationship between infestation and wet weight of invertebrates was specified using a smoothed cubic regression spline with $k = 5$ knots. A smoothed random effect term for each specimen nested within the mussel bed was specified to account for repeated shell temperature measurements on the same individual at different time intervals while allowing intercepts to vary between mussel beds. Model fits were assessed using residual diagnostics using the 'appraise' function from the 'gratia' R package [61]. To test for a significant effect of infestation on shell temperature or wet weight, parametric $F$-tests were computed ($p < 0.05$) using the 'anova.gam' function from the 'mgcv' R package [62]. Wald's-like tests were used to assess whether the non-linear relationship between infestation and shell temperature over time (i.e., heating rates) differed between infested vs. non-infested infauna and whether the non-linear relationship between infestation and wet weight of invertebrates over time (i.e., desiccation rates) differed between infaunal species [63].

## 3. Results

*3.1. Body Temperature Assessment*

A general trend was observed among all experimental dates for biomimetic mussels. Regardless of the date or the time interval considered, infested biomimetic mussels exhibited lower shell temperatures (1.7–4.2 °C lower) than non-infested mussels ($p < 0.001$, Table S1, Figure 1). Shell temperatures of biomimetic mussels increased with time, but mostly in the first 30 min, with infested mussels warming at a slower rate than non-infested mussels ($p < 0.001$, Table S2, Figure 1). By the end of the experiment, shell temperatures in non-infested and infested mussel beds had increased by 70.9 and 63.3%, respectively. Variations in mussel shell temperatures and warming rates were observed between dates and were linked to variations in environmental conditions, especially in UV index and wind speed ($p < 0.05$, Tables 1 and S3, Figure 1).

For *Acanthochitona garnoti* and *Scutellastra granularis*, shell temperatures (averaged over all the time periods) were significantly lower on infested mussel beds than on non-infested mussel beds ($p < 0.001$, Tables S4 and S8, Figure 2). Although heating rates of *A. garnoti* were not statistically different (Table S5, Figure 2), heating rates of *S. granularis* were significantly slower on infested mussel beds than on non-infested mussel beds ($p < 0.05$, Table S9, Figure 2). Mollusc shell temperatures and heating rates were not significantly different between experimental dates for these two species (Tables 1, S6 and S10, Figure 2). Finally, for both *A. garnoti* and *S. granularis*, the shell temperatures of molluscs were always lower than the shell temperatures of the biomimetic mussels for each corresponding infestation condition and experimental date ($p < 0.001$, Tables S7 and S11, Figure 2).

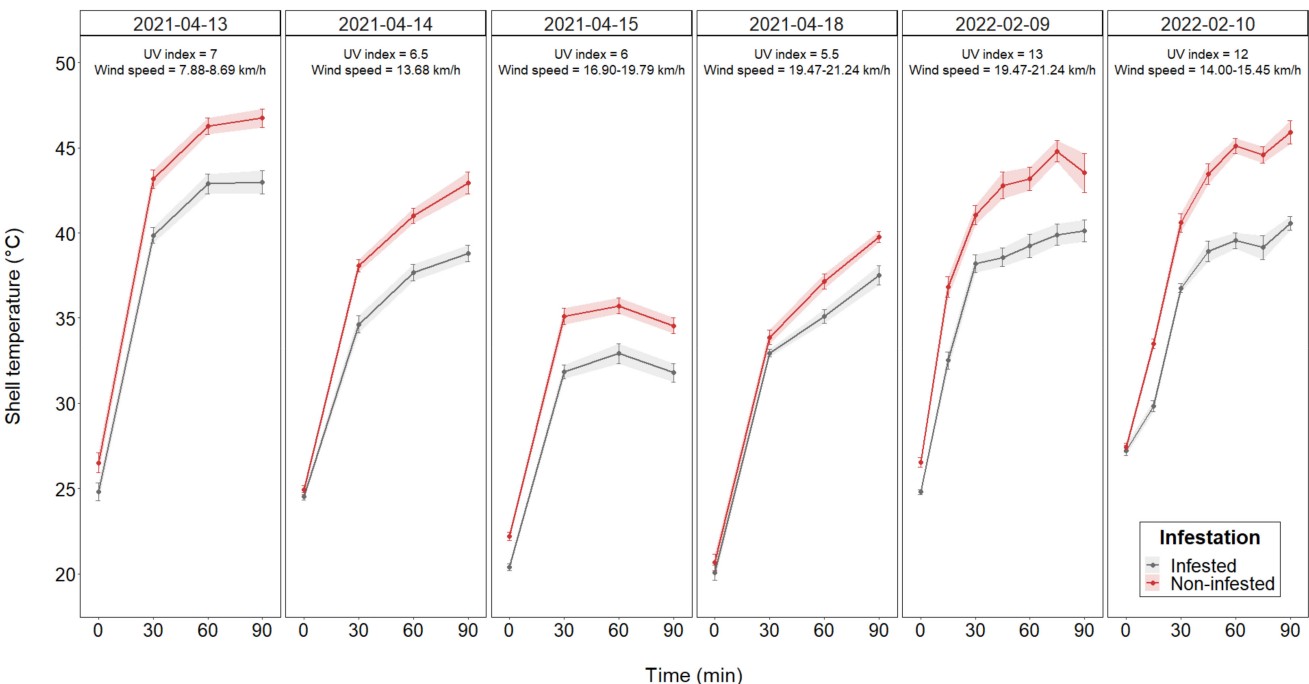

**Figure 1.** Changes in mean (± SE) shell temperatures (in °C) of infested and non-infested individual biomimetic mussels (*Perna perna*) for 90 min after emersion (*n* = 3 mussel beds) on each experimental date.

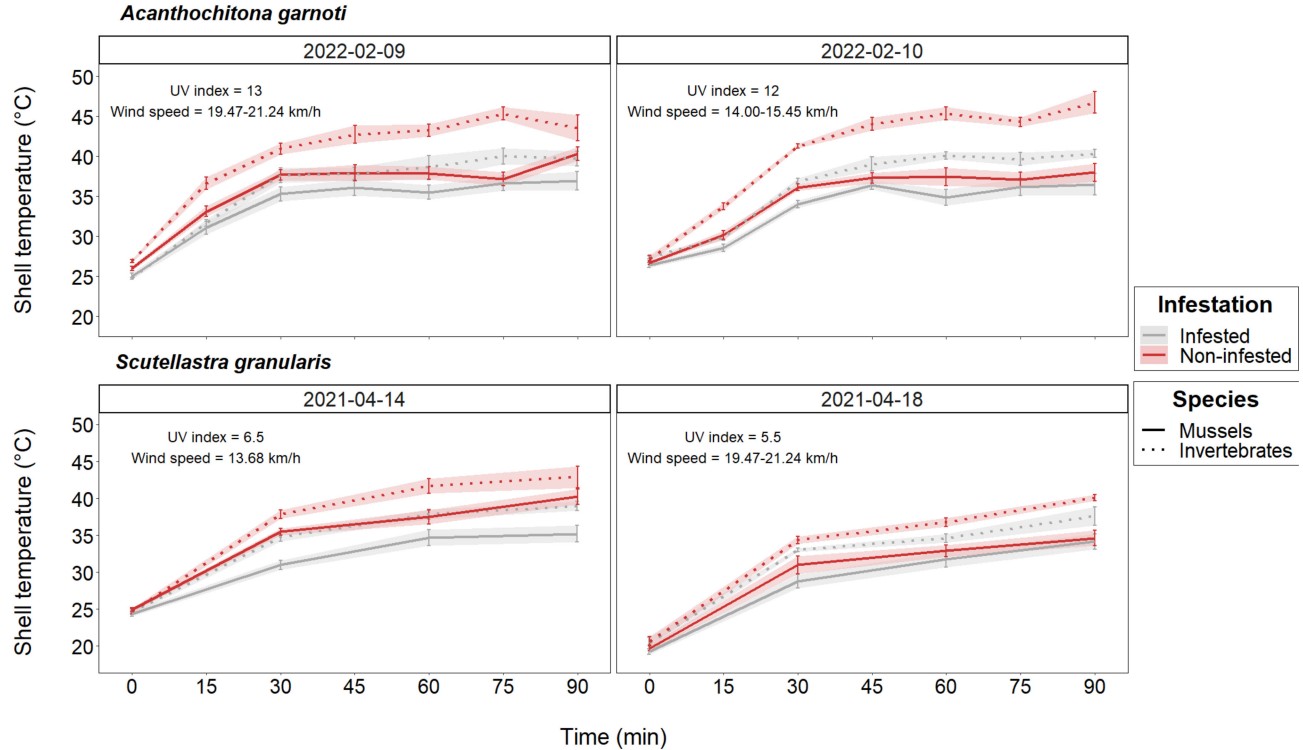

**Figure 2.** Changes in mean (± SE) shell temperatures (in °C) of the chiton (*Acanthochitona garnoti*) and the limpet (*Scutellastra granularis*), with the corresponding biomimetic mussels (*Perna perna*), in either infested or non-infested mussel beds (*n* = 3) for 90 min after emersion, on each corresponding experimental date.

For *Oxystele antoni* and *Helcion pectunculus*, their shell temperatures and those of biomimetic mussels, as well as their respective warming rates, were never significantly

different between species or infestation status of the mussel bed (Tables S12 and S13, Figure 3). Variations in mollusc and biomimetic mussel shell temperatures were observed between experimental dates ($p < 0.05$, Tables 1 and S14, Figure 3) and were linked to variations in environmental conditions, especially in wind speed.

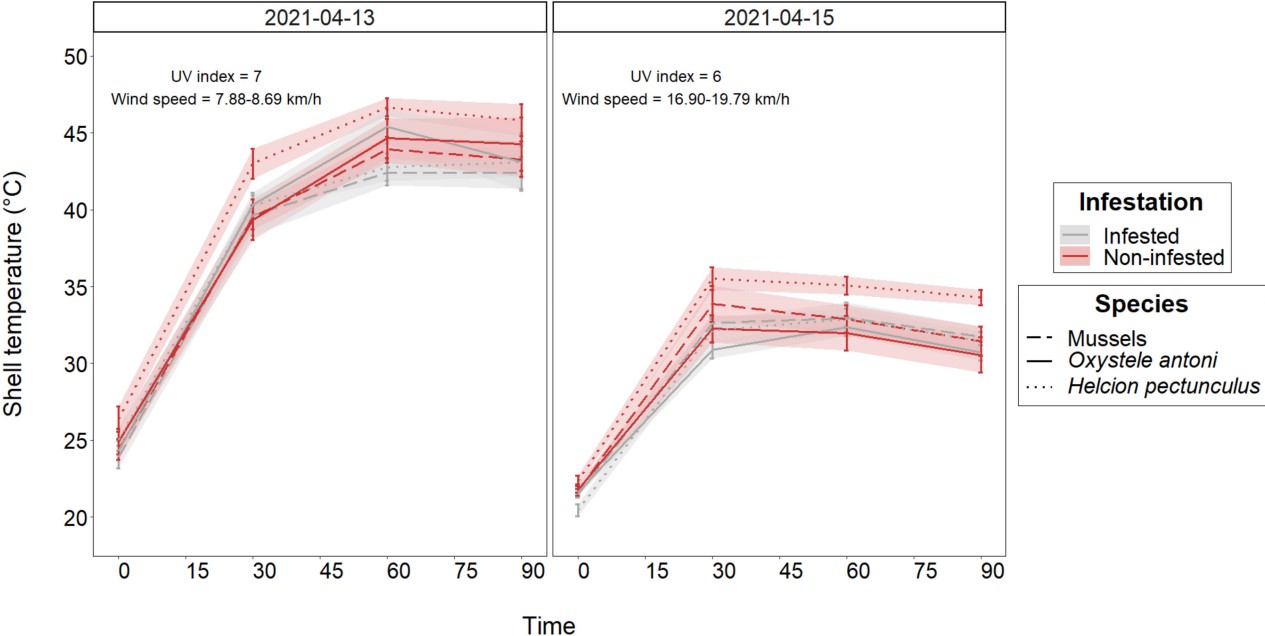

**Figure 3.** Changes in mean ($\pm$ SE) shell temperatures (in °C) of the limpet (*Helcion pectunculus*), the top shell (*Oxystele antoni*), and biomimetic mussels (*Perna perna*) in either infested or non-infested mussel beds ($n = 3$) for 90 min after emersion, on each experimental date.

### 3.2. Dessication Assessment

No statistical differences in wet weight or desiccation rates of invertebrate specimens were detected between infested and non-infested mussel beds (Table S15). Variations in wet weight, but not in desiccation rates (Table S16), were detected between invertebrate species, regardless of infestation levels of the mussels ($p < 0.05$, Table S15).

### 4. Discussion

Rocky shores are amongst the most physically stressful environments on the planet. Despite the severity of physical gradients, mainly temperature and desiccation, rocky intertidal shores are biologically rich ecosystems. Global climate change (GCC) is likely to be associated with an increase in the physical gradients observed on rocky shores and an increased frequency of heat waves [5,7]. These changes will likely cause shifts in community structure and local extinctions [11–13]. However, under stressful conditions, intertidal organisms can display thermal behavioural adaptations, such as seeking refuge in mussel beds, which mitigate temperature and desiccation stress [21–24]. Numerous studies have investigated the effects of infestation by euendolithic shell-boring parasites on individual mussels and the mussel bed microclimate (reviewed in [38]). To our knowledge, this study is amongst the first attempts to assess the indirect thermodynamic benefits conferred by euendolithic infestation of mussel beds to their associated infauna [64].

The results of our manipulative experiments indicated a buffering role of photoautotrophic shell-degrading euendoliths on the thermal stress experienced by mussels. Infested biomimetic mussels displayed significantly lower shell temperatures and slower warming rates than non-infested biomimetic mussels, even when environmental conditions were not considered stressful (i.e., low solar irradiance and high wind speed, Figure 1). Since biomimetic mussels have been extensively used to record organismal body temperatures [41,58,65,66] and avoid potential confounding factors (e.g., gaping behaviour,

architectural complexity), our results effectively isolate a marked cooling effect of euendolithic infestation through the whitening of the mussel shell and of the mussel bed. This is in agreement with previous studies conducted with both live and biomimetic mussels with and without euendolithic infestation [31,34,41,42,67].

Euendolithic infestation of mussel shells (i.e., substratum), at times, indirectly influenced the body temperatures of associated invertebrates but not their warming rates. In our study, the light-coloured invertebrates (i.e., *Acanthochitona garnoti* and *Scutellastra granularis*) displayed lower shell temperatures on euendolith-infested mussels compared to non-infested mussels and were always cooler than the substratum. In Portugal, Zardi et al. obtained similar results for the limpet *Patella vulgata* (Linnaeus, 1758), the snail *Littorina littorea* (Linnaeus, 1758), and mussel recruits using artificial mussel beds made of blue mussels, *Mytilus galloprovincialis* (Lamarck, 1819) [64]. However, this was not the case for the dark-coloured invertebrates in this study (i.e., *Helcion pectunculus* and *Oxystele antoni*), for which shell temperatures did not differ significantly between infestation status and with the substratum. Body temperatures of intertidal ectotherms (including mussels) are primarily driven by non-climatic heat sources, either directly through solar irradiance or indirectly through re-radiation by the substratum [48,52]. The latter is especially true for organisms with a large foot that maintains conducive contact with their substrate, such as limpets, snails, or chitons [46,48,50,52,53,68]. Consequently, molluscs gain considerable heat from the substratum (e.g., [68]), often exhibiting the same temperature as the substratum [48,50], though they can sometimes become even warmer [21].

At a finer scale, the organism's morphology, such as the shell colour and shape, or behaviour, may also influence its thermal properties [22,51,68,69]. This helps explain why euendolithic effects on the temperatures of mussels indirectly affect the temperatures of some animals but not others. Darker organisms absorb more heat from solar irradiance and do so faster [70] but also lose heat faster through convection [71]. Taller shells, with a more circular aperture, have a smaller contact area with the substratum, reducing heat gain through conduction [51,53,72], and, being taller, project into faster wind velocities, increasing heat loss through convection [69]. Although the wind has a cooling effect on the mussels themselves [30], because of the architectural complexity and intricate matrix of byssal threads within mussel beds, the effects of wind on the infauna would be negligible. Highly sculpted shells displaying heavy ridges, bumps, or other features, have a larger contact area with the atmosphere, thereby increasing heat loss through convection. This process is, however, only effective at high wind speeds [51,69] and, again, is not directly important within a mussel bed. In addition, intertidal organisms can display a variety of behavioural adaptations to thermal and desiccation stress, including seeking thermal refuges (e.g., the underside of rocks, crevices, and mussel beds) for most invertebrates [23,24], 'mushrooming' for limpets [21], or forming aggregations for snails [47].

We selected four intertidal invertebrates known to inhabit mussel beds commonly. On the one hand, the morphological differences between mollusc species could, in part, explain why an indirect effect of euendolithic infestation on body temperatures was only detected for *Acanthochitona garnoti* and *Scutellastra granularis*, for which heat transfer through conduction from the substratum (here, the mussel bed) is maximized. Indeed, *A. garnoti* and *S. granularis* (heavily eroded) are light in colour, with a large foot and little to no shell features, whereas *Helcion pectunculus* and *Oxystele antoni* are both dark in colour, the former having a heavily sculpted shell, while the latter has a coiled smooth shell. On the other hand, behavioural differences in the face of thermal and desiccation stress could have influenced the body temperatures of invertebrates. Most invertebrate specimens explored the substratum for a short time, between 5 and 10 min after the beginning of the experiments. Afterward, some specimens displayed thermoregulatory behaviours, such as seeking thermal refuges within the mussel beds for *A. garnoti* [73], 'mushrooming' for the two limpet species (i.e., *S. granularis* and *H. pectunculus*) [21], or adopting a standing position for *O. antoni* [74]. However, this was not the case for all invertebrate specimens. Thermoregulatory behaviours were thus only sporadically observed during our experiment

in both mussel bed infestation treatments. Moreover, invertebrates seeking refuge between or under the biomimetic mussels could not be captured by the infrared camera and were not integrated into this study. Finally, thermoregulatory behaviours are often effective in the short term and represent high-risk strategies for invertebrates [21,74].

Our desiccation stress experiments did not detect an effect of euendolithic infestation of the mussel bed on water loss by infaunal invertebrates. This could be explained by the short exposure time of invertebrates to adverse environmental conditions (i.e., 90 min), the non-stressful conditions on the experimental dates (i.e., low solar irradiance and medium to high wind speeds), and the various behaviours displayed by invertebrates to minimize water loss under stress. It is also noted that the wet weight of each specimen used was not accounted for at the start of the experiment. Replicating the experiments while taking into account the initial wet weight, within species and between species, could potentially help to assess, with more accuracy, water loss.

## 5. Conclusions

We show that euendolith-induced corrosion enhances the quality of mussel beds as thermal refugia for selected invertebrate species. The beneficial effects of euendolithic corrosion on the mussel bed as a habitat are, however, highly variable at fine scales of space and time, which is true for the intertidal as a whole. Moreover, the colour phenotype of the species investigated is important in determining whether it benefitted from the indirect cooling effects of euendoliths. Thus, the additional thermal buffering provided by euendolithic infestation of mussel shells is only relevant if the organisms are thermally stressed, and its extent depends on the species in question. Under GCC, intertidal ecosystems are expected to greater suffering from extreme temperature and desiccation stress in the future, while euendolithic infestation of mussels will become more severe. In this context, euendolithic infestation may improve the chances of survival of mussels and the quality of mussel beds as thermal refuges for associated invertebrates, which could, in turn, decrease their susceptibility to local extinctions. At the same time, euendolithic infestation also has negative impacts that could hinder the quality of habitat offered by mussel beds or its stability through time [38]. In addition, the bioerosive activities of photoautotrophic euendoliths and grazers can interact and result in increased levels of shell erosion [75]. It is thus important to understand the ecological consequences of euendolithic infestation on mussel beds as a habitat and how this parasitic/mutualistic relationship will influence the wider ecosystem, that is, the rocky shore.

**Supplementary Materials:** The following supporting information can be downloaded at: https://www.mdpi.com/article/10.3390/d15020239/s1, Table S1: Summary outputs of the Generalized Additive Model (GAM) on the effect of euendolithic infestation on biomimetic mussel shell temperatures (data pooled for all experimental dates). The model used in this analysis: shell_temp ~ infestation + s(time, bs = "cr," by = infestation, k = 4) + s(id, bs = "re"); Table S2: Results of Wald's-like test to assess whether the non-linear relationship between infestation and biomimetic mussel shell temperatures differ over time (data pooled for all experimental dates); Table S3: Results of Wald's-like test to assess whether the non-linear relationship between infestation and biomimetic mussel shell temperatures over time differ between experimental dates; Table S4: Summary outputs of the Generalized Additive Model (GAM) on the effect of euendolithic infestation of the biomimetic mussels on the shell temperatures of the spiny chiton, *Acanthochitona garnoti* (data pooled for all experimental dates). The model used in this analysis: shell_temp infestation + s(time, bs = "cr," by = infestation, k = 6) + s(id, bs = "re"); Table S5: Results of Wald's-like test to assess whether the non-linear relationship between infestation and shell temperatures of the spiny chiton, *Acanthochitona garnoti*, differ over time (data pooled for all experimental dates); Table S6: Results of Wald's-like test to assess whether the non-linear relationship between infestation and shell temperatures of the spiny chiton, *Acanthochitona garnoti*, over time differ between experimental dates; Table S7: Summary outputs of the Generalized Additive Model (GAM) on the effect of euendolithic infestation on the biomimetic mussel shell temperatures and the shell temperatures of the spiny chiton, *Acanthochitona garnoti* (data pooled for all experimental dates). The model used in this analysis: shell_temp ~ infestation + species + s(time, bs = "cr,"

by = infestation, k = 6) + s(id, bs = "re"); Table S8: Summary outputs of the Generalized Additive Model (GAM) on the effect of euendolithic infestation of the biomimetic mussels on the shell temperatures of the granular limpet, *Scutellastra granularis* (data pooled for all experimental dates). The model used in this analysis: shell_temp ~ infestation + s(time, bs = "cr," by = infestation, k = 4) + s(id, bs = "re"); Table S9: Results of Wald's-like test to assess whether the non-linear relationship between infestation and shell temperatures of the granular limpet, *Scutellastra granularis*, differ over time (data pooled for all experimental dates); Table S10: Results of Wald's-like test to assess whether the non-linear relationship between infestation and shell temperatures of the granular limpet, *Scutellastra granularis*, over time differ between experimental dates; Table S11: Summary outputs of the Generalized Additive Model (GAM) on the effect of euendolithic infestation on the biomimetic mussel shell temperatures and the shell temperatures of the granular limpet, *Scutellastra granularis* (data pooled for all experimental dates). Model used in this analysis: shell_temp ~ infestation + species + s(time, bs = "cr", by = infestation, k = 4) + s(id, bs = "re"). Table S12: Summary outputs of the Generalized Additive Model (GAM) on the effect of euendolithic infestation on the biomimetic mussels and on the shell temperatures of the prickly limpet, *Helcion pectunculus*, and the variegated top shell, *Oxystele antoni* (data pooled for all experimental dates). The model used in this analysis: shell_temp ~ infestation + s(time, bs = "cr," by = infestation, k = 4) + s(id, bs = "re"); Table S13: Results of Wald's-like test to assess whether the non-linear relationship between infestation and shell temperatures of the prickly limpet, Helcion pectunculus, and the variegated top shell, Oxystele antoni, differ over time (data pooled for all experimental dates); Table S14: Results of Wald's-like test to assess whether the non-linear relationship between infestation and shell temperatures of the prickly limpet, Helcion pectunculus, and the variegated top shell, Oxystele antoni, over time differ between experimental dates; Table S15: Summary outputs of the Generalized Additive Model (GAM) on the effect of euendolithic infestation on the wet weight of infaunal invertebrates (data pooled for all species and experimental dates). The model used in this analysis: log(wet_wgt) ~ infestation * species + s(time, bs = "cr," by = infestation, k = 5) + s(id, bs = "re"); Table S16: Results of Wald's-like test to assess whether the non-linear relationship between infestation and wet weight of infaunal invertebrates (data pooled for all species and experimental dates) differ over time between species (data pooled for all experimental dates).

**Author Contributions:** Conceptualization, G.I.Z., K.R.N. and A.M.D.; methodology, C.D.M., G.I.Z., K.R.N. and A.M.D.; fieldwork, A.M.D.; formal analysis, A.M.D.; data collection, A.M.D.; data curation, A.M.D.; writing—original draft preparation, A.M.D.; writing—review and editing, A.M.D., C.D.M., G.I.Z., K.R.N. and P.W.F.; visualization, A.M.D.; supervision, P.W.F., C.D.M. and G.I.Z.; project administration, A.M.D.; funding acquisition, P.W.F. and C.D.M. All authors have read and agreed to the published version of the manuscript.

**Funding:** This study was funded by the National Research Foundation (NRF) of South Africa, grant number 64801, and further supported by ANR SAN22202 to K.R.N. This project has received funding from the European Union's Horizon 2020 Research and Innovation programme under the Marie Skłodowska Curie, grant agreement No. 101034329. G.I.Z. is the recipient of the WINNINGNormandy Program supported by the Normandy Region.

**Institutional Review Board Statement:** All the necessary permits for the collection of the invertebrates were obtained from the Department of Agriculture, Forestry and Fisheries (DAFF), Republic of South Africa (permit reference number: RES2018/46).

**Data Availability Statement:** Infrared pictures, data, and R scripts are available online on GitHub (https://github.com/AlexiaDievart/hot_invertebrates, accessed on 12 December 2022).

**Acknowledgments:** Fieldwork, Tristan A. van Rooyen; formal analysis and writing—review and editing, Guy F. Sutton. The authors would also like to acknowledge Rhodes University for providing facilities to conduct the study.

**Conflicts of Interest:** The authors declare no conflict of interest. The funders had no role in the design of the study; in the collection, analyses, or interpretation of data; in the writing of the manuscript; or in the decision to publish the results.

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
