# Peer review of "Euendolithic Infestation of Mussel Shells Indirectly Improves the Thermal Buffering Offered by Mussel Beds to Associated Molluscs, but One Size Does Not Fit All"

_diversity, doi:10.3390/d15020239_

Round 1
Reviewer 1 Report
The authors analyze the effects of an understudied group of photoautotrophic euendolites on the thermal biology of tidal mollusks. Based on a series of well-planned experiments, various focal species, and correct mathematical analyzing of the data, this very interesting manuscript details the hypothesis and results and is well discussed. Very good job!
Author Response
We are glad you enjoyed the manuscript ! Thank you very much for your review.
Reviewer 2 Report
The manuscript analyzes the effect of euendolithic infestation in mussel beds in provide indirect thermorregulary effect in selected species of marine molluscs. The results shows that this effect exist depending on the species, and euendolithic infestation for invertebrate communities associated with mussel beds may mitigate intertidal local extinction events triggered by marine heatwaves. The paper is well written and analyzed. I have only some minor issues that do not change the content of the work.
Specific comments
The version I received had some comments from different authors. I ask you for corresponding author check if the version she submitted is the last one of this manuscript.
Page 1, lines 38-40: I think the second sentence of the introduction could the be first one.
Page3, line 120: According to zoological nomenclature, The author of Oxystele antoni should appear without blackets, because the name was originally proposed in the genus Oxystele.
Page 4, line 165: Infaunal species are species that live buried in soft substrate (e.g. sand, mud). Considering the species you studied, I think this is not the correct term to describe them.
Page 4, line 179: Considering you also sampled the temperature in different times, I think it is more correct use a Generalized Additive Mixed Models adjusted to a repeated measure design to analyze the temperature change throughout the time, putting time as other fixed effect.
Page 5, lines 212-213: The previous comments of mixed effect models made in line 179 also applies here.
Author Response
Thank you for your review. Here is the point-by-point response to your comments:
Point 1: The version I received had some comments from different authors. I ask you for corresponding author check if the version she submitted is the last one of this manuscript.
Response 1: A version without authors comments was submitted by e-mails to the editors of Diversity. Sorry for this oversight.
Point 2: Page 1, lines 38-40: I think the second sentence of the introduction could be the first one.
Response 2: I agree, it does flow better, so change was done. For the editors, this changes the order of the references throughout the manuscript.
Point 3: Page3, line 120: According to zoological nomenclature, The author of Oxystele antoni should appear without brackets, because the name was originally proposed in the genus Oxystele.
Response 3: Thanks for removing the doubt here.
Point 4: Page 4, line 165: Infaunal species are species that live buried in soft substrate (e.g. sand, mud). Considering the species you studied, I think this is not the correct term to describe them.
Response 4: Confusing, thus removed.
Point 5: Page 4, line 179: Considering you also sampled the temperature in different times, I think it is more correct use a Generalized Additive Mixed Models adjusted to a repeated measure design to analyze the temperature change throughout the time, putting time as other fixed effect.
Page 5, lines 212-213: The previous comments of mixed effect models made in line 179 also applies here.
Response 5: Generalized additive mixed models (GAMMs) are an extension of generalized additive models incorporating random effects, while you suggest to add 'time' as a fixed effect, which is a bit confusing. Our models incorporate the effect of time on temperature, which is non-linear, and allow it to vary between euendolithic infestation levels. This basically allows the pieces between time intervals to hinge. These models captured the best the structure of our data, so were selected in consequence. However, this concern will be investigated.
Reviewer 3 Report
This manuscript describes an interesting set of experiments conducted to determine the effect of mussel discoloration (whitening) on other molluscs living interstitially in the mussel bed. The authors demonstrated that of the four molluscs (two limpets, a top shell and a chiton), a light coloured limpet and a light coloured chiton kept their cool amongst the whitened mussel shells. In contrast the shell temperatures of the dark coloured limpet and top shell were about the same with the whitened mussels.
I have no major criticisms. However, it would be instructive if the authors could provide a brief introduction to ‘euendolithic infestation’ in the Introduction section. As the experiments have demonstrated that the four molluscan species warm up differently in controlled conditions, perhaps the authors could speculate, apart from their surface colouration, whether their movements or behaviour could influence the results obtained in the Discussion section. Over the longer term, do these grazers affect the extent of euendolithic infestation on the mussels?
Other comments are indicated in the attached pdf.

Author Response
Thank you for your comments on the manuscript ! Please find attached the PDF on which you commented, with the response to your comments.
To answer some of your points here:
- Information on what are photoautotrophic euendoliths and where to find them have been provided in the introduction.
- A paragraph on the influence of behaviour and/or movement of invertebrates on the conclusion of the thermal experiment was added in the discussion
- The caveat in the method of the desiccation experiment was addressed in the corresponding paragraph in the discussion.
Finally, to answer your last question : Yes, there could be an interactive effect of euendoliths and grazers, that would result in increased levels of bioerosion on the mussel shell. This has not been demonstrated experimentally however. I added a short sentence on this subject in the conclusion, as this reflexion is outside the scope of this research manuscript. I discussed this in more details in the following review: https://www.mdpi.com/1424-2818/14/9/737

Reviewer 4 Report
It is a rather short communication, containing a piece of novel and interesting information which can be potentially useful for marine ecologists. There are no serious shortcomings and errors in the text, however, before acceptance, I would recommend the authors to make some improvements to the original text. See my specific comments below.
Resume (lines 16-167) “Mussel shells are 16 often parasitized by photoautotrophic euendoliths” – are you sure that these euendoliths are true parasites, not commensals? I doubt that the term “parasitism” is relevant here (the benefitial effect of these microorganisms on mussels does, rather, fit to the term “mutualism”). Please, explain this, or, at least, add some phrases about their parasitic relationships with the mussels to “Introduction”.’
Line 127 “on rocky shores on the South and East coasts”. On coasts of what? #
What is the place of origin of the specimens of the infaunal mollusks used in experiments? How they were collected? Were they subjected to some acclimation procedure? What the thermal regime of their holding tanks was? In other words, are you sure that the condition of these animals was equal to the conditions of their conspecific in natural environment? All these details are missing from the ‘M&Ss’ section.
Paragraphs 2.2 and 2.3 contain many identical phrases. Is it possible to avoid repetitions, for example, by uniting the description of the same statistical procedure?
Lines 252-253 “Finally, for both A. garnoti and S. granularis, the shell temperatures of invertebrates”. All species used in experiments, both mussels and non-mussels, are mollusks. I would recommend to replace word “invertebrates” with “mollusks” in this context. The term “invertebrate” is too broad and covers not only intertidal benthic organisms but also a plethora of swimming and burrowing forms (for example, medusae, polychaetes, planktonic animals). The same is applicable to line 261 (“For Oxystele antoni and Helcion pectunculus, shell temperatures of invertebrates…” – why not to write “of these mollusks”?).
Lines 302-303 “This is in agreement with previous studies conducted with both live and 302 biomimetic mussels with and without euendolithic infestation”. In that case, what is the novelty of your research?
Line 312-314 “However, this was not the case for the dark-coloured invertebrates in this study (i.e., Helcion pectunculus and Oxystele antoni), for which shell temperatures did not differ significantly between infestation status and with the substratum”. The “M&Ms” section lacks even a shorter description of biological and morphological characters of the studied mollusks, which may be relevant in the context. For example, an important difference between the light- and dark-coloured mollusks appears only in “Discussion”, whereas it would be much more desirable to see this information in the “M&Ms” chapter. I recommend to place the content of lines 345-352 to the description of the study design.
Lines 325 and 326 “organism’s phenotype, such as the shell colour, morphology or behaviour may also influence its thermal properties”. There is a mess in this phrase. Behaviour is NOT a part of a phenotype, shell colour is a part of morphology (in the broad sense) etc. Please, be careful with the using scientific terms with exact meaning!
Author Response
Thank you for your comments on manuscript ! Here is a point-by-point answer:
It is a rather short communication, containing a piece of novel and interesting information which can be potentially useful for marine ecologists. There are no serious shortcomings and errors in the text, however, before acceptance, I would recommend the authors to make some improvements to the original text. See my specific comments below.
Point 1: Resume (lines 16-167) “Mussel shells are 16 often parasitized by photoautotrophic euendoliths” – are you sure that these euendoliths are true parasites, not commensals? I doubt that the term “parasitism” is relevant here (the benefitial effect of these microorganisms on mussels does, rather, fit to the term “mutualism”). Please, explain this, or, at least, add some phrases about their parasitic relationships with the mussels to “Introduction”.’
Response 1: Photoautotrophic euendoliths ultimately cause the death of the mussel, by weakening the shell so much that the mussels break it themselves when trying to close it. The beneficial effect of euendolithic infestation is an unvoluntary by-product of their erosive activity, which causes the whitening of the mussel shell and thus increases its albedo. This beneficial effect is restricted to particular environmental conditions, when solar irradiance is high and wind is low. I agree that this is a tricky relationship to define, and it has been coined parasitic, mutualistic, and even symbiotic in other scientific papers.
The following changes were made (see underlined, line numbers have changed):
- Abstract:
Mussel shells are often colonized by photoautotrophic euendoliths, which have indirect conditional beneficial thermoregulatory effects on both solitary and aggregated mussels by increasing the albedo of the shell.
- Introduction
Besides its negative sub-lethal and lethal effects to mussels [38–40], euendolithic infestation causes a distinctive discolouration of the mussel shell as a by-product of its corrosive activity, thereby increasing its albedo [33,34].
Point 2: Line 127 “on rocky shores on the South and East coasts”. On coasts of what? #’
Response 2: Of South Africa, added.
Point 3: What is the place of origin of the specimens of the infaunal mollusks used in experiments? How they were collected? Were they subjected to some acclimation procedure? What the thermal regime of their holding tanks was? In other words, are you sure that the condition of these animals was equal to the conditions of their conspecific in natural environment? All these details are missing from the ‘M&Ss’ section.
Response 3: Information added in the M&M.
Point 4: Paragraphs 2.2 and 2.3 contain many identical phrases. Is it possible to avoid repetitions, for example, by uniting the description of the same statistical procedure?
Response 4: Indeed. Information on the statistical procedure was reworked into an additional paragraph.
Point 5: Lines 252-253 “Finally, for both A. garnoti and S. granularis, the shell temperatures of invertebrates”. All species used in experiments, both mussels and non-mussels, are mollusks. I would recommend to replace word “invertebrates” with “mollusks” in this context. The term “invertebrate” is too broad and covers not only intertidal benthic organisms but also a plethora of swimming and burrowing forms (for example, medusae, polychaetes, planktonic animals). The same is applicable to line 261 (“For Oxystele antoni and Helcion pectunculus, shell temperatures of invertebrates…” – why not to write “of these mollusks”?).
Response 5: Changes were made throughout the manuscript.
Point 6: Lines 302-303 “This is in agreement with previous studies conducted with both live and 302 biomimetic mussels with and without euendolithic infestation”. In that case, what is the novelty of your research?
Response 6: There might be a misunderstanding here. The fact that euendolith-infested mussels display lower temperatures than non-infested ones needed to be confirmed in our study. However, this was not the focus of the study. The aim of our study was to investigate if the thermal buffering effect of euendolithic infestation of mussel shells (confirmed and detected during our experiments) would reflect in lower body temperatures for associated molluscs (which was the case for some species but not others, and dependent on the environmental conditions).
Point 7: Line 312-314 “However, this was not the case for the dark-coloured invertebrates in this study (i.e., Helcion pectunculus and Oxystele antoni), for which shell temperatures did not differ significantly between infestation status and with the substratum”. The “M&Ms” section lacks even a shorter description of biological and morphological characters of the studied mollusks, which may be relevant in the context. For example, an important difference between the light- and dark-coloured mollusks appears only in “Discussion”, whereas it would be much more desirable to see this information in the “M&Ms” chapter. I recommend to place the content of lines 345-352 to the description of the study design.
Response 7: Added in the M&M. Paragraph in the discussion was modified in consequence.
Point 8: Lines 325 and 326 “organism’s phenotype, such as the shell colour, morphology or behaviour may also influence its thermal properties”. There is a mess in this phrase. Behaviour is NOT a part of a phenotype, shell colour is a part of morphology (in the broad sense) etc. Please, be careful with the using scientific terms with exact meaning!
Response 7: Sentence was modified. Phenotype often include behaviour in many research papers and definitions found online (e.g., https://www.britannica.com/science/phenotype). In order to remove any doubt or misunderstanding, it would be useful (and educative at a personal level) to provide the references to support your comment.